# Two New *Ferula* (Apiaceae) Species from Central Anatolia: *Ferula turcica* and *Ferula latialata*

Hüseyin Onur Tuncay [1,2,*], Emine Akalın [1,3], Aslı Doğru-Koca [4], Fatma Memnune Eruçar [5,6] and Mahmut Miski [5]

1. Department of Pharmaceutical Botany, Faculty of Pharmacy, Istanbul University, Istanbul 34116, Türkiye
2. Department of Pharmaceutical Botany, Institute of Graduate Studies in Health Sciences, Istanbul University, Istanbul 34116, Türkiye
3. Faculty of Pharmacy, Eastern Mediterranean University, Famagusta 99628, Türkiye
4. Department of Biology, Faculty of Science, Hacettepe University, Ankara 06800, Türkiye
5. Department of Pharmacognosy, Faculty of Pharmacy, Istanbul University, Istanbul 34116, Türkiye
6. Department of Pharmacognosy, Institute of Graduate Studies in Health Sciences, Istanbul University, Istanbul 34116, Türkiye
* Correspondence: onur.tuncay@istanbul.edu.tr

**Abstract:** *Ferula turcica* and *Ferula latialata* are two novel endemic species discovered in the Konya and Kırşehir provinces of the central Anatolian region of Türkiye. These two new species are described by morphological, ecological, carpological, and phytochemical characteristics and phylogenetic analysis. *F. turcica* and *F. latialata* are morphologically distinct from *F. szowitsiana* by their habit, the stalk of the terminal umbella, and the mericarp size, as well as by the profile of their secondary metabolite markers and phylogenetic placement. The phylogenetic analyses of sequences of the internal transcribed spacer in ribosomal DNA belonging to both new taxa were conducted to reveal the evolutionary relationships of the new species. Their relationships with the other related species and proposed conservation status were reviewed. The morphological, molecular, and phytochemical evidence supported the hypothesis that *Ferula turcica* and *Ferula latialata* are two new distinct species.

**Keywords:** *Ferula*; new species; Turkey; Apiaceae; morphology; anatomy; chemotaxonomy; molecular; phylogeny

## 1. Introduction

The Apiaceae family is one of the largest families among Angiosperm plants [1,2]. *Ferula* L. is the largest genus in the Apiaceae family, with approximately 213 species [3]. *Ferula* species are widespread in the temperate regions of the Euro-Asian continent, surrounded by the Canary Islands in the West, North Africa in the South, China and India in the East, and Central Europe in the North.

*Ferula* plants have been used for medicinal and culinary purposes since ancient times [4,5]. Pedanius Dioscorides described the medicinal properties of several *Ferula* resins, including asafoetida (*Ferula assa-foetida* L.), galbanum (*F. gummosa* Boiss.), sagapenum (*F. persica* Willd.), and African ammoniacum (*F. marmarica* Asch. and Taub.), in his De Materia Medica two thousand years ago [6]. Ibn-i Sina (Avicenna) described the application of the oleo-gum-resin from *Ferula foetida* in the treatment of cancerous tumors in the Canon of Medicine [7]. The resins of *Ferula* species have been used in the food and health industries as a spice, nutraceutical, and cosmeceutical in India, Iran, and Afghanistan [8,9]. Antimicrobial and anti-inflammatory activities of the essential oil and extracts of the aerial parts of *F. szowitsiana* have been reported [10,11]. The roots of *F. persica* have been used to alleviate the symptoms of diabetes in Iran and Jordan [12].

Many taxonomic studies have been conducted on the genus *Ferula*. Boissier classified the genus *Ferula* species grown in the Irano-Turanian region into three sections based on

the shape of their petals and the number of their vittae: *Peucedanoides* Boiss. *Scrodosma* Bunge and *Euferula* Boiss. [13].

Korovin established the most comprehensive infrageneric classification by defining the *Ferula* genus into six subgenera and eight divisions [14]. Conversely, in the study conducted by Safina and Pimenov, *Ferula* species were examined in terms of their carpological characters, and they emphasized that the species of the *Ferula* subgenus did not form a homogeneous group [15]. A molecular study conducted by Pimenov et al. on 90 *Ferula* species yielded quite different results in comparison with Korovin's taxonomical classification [16].

According to Flora of Turkey and the East Aegean Islands, 18 *Ferula* species were listed by Peşmen in Türkiye [17]. Afterward, Sağıroğlu and Duman found four new species as a result of their revised study on *Ferula* species growing in Türkiye [18–21]. Subsequently, *F. divaricata* Pimenov and Kljuykov and *F. pisidica* Akalın and Miski were discovered in 2013 and 2020, respectively [22,23]. Followed by the addition of these novel species, the total number of *Ferula* species growing in Türkiye has reached 24.

Although based on its morphological affinity, the genus *Ferula* was accepted as a member of the tribe Peucedaneae, its classification was updated and transferred to the tribe Scandiceae based on the phylogenetic hypothesis [24]. The intrageneric phylogenetic relationships of *Ferula* are ongoing [25–28]. The sequences of the internal transcribed spacers (ITS 1, 5.8S rRNA, ITS 2) were one of the markers used in these studies. According to Panahi et al., the *Ferula* species distributed in the southwestern part of the Iran-Turanian floristic region generate a monophyletic clade and additional polytomic *F. narthex*. These lineages correspond to subgenus *Narthex* (Falc.), section *Merwia* (B. Fedtsch.) Korovin. Their results suggested that the interspecific boundaries in this section are unclear. Especially the ITS sequences of some species showed that they are identical to each other and are very close species in terms of their morphological features or were determined as synonyms. For instance, the ITS sequences of *F. gummosa* are identical to those of *F. badrakema*, *F. linczevskii*, *F. undulata*, and *F. myrioloba*.

Additionally, because of the disagreement between hypothetical phylogenetic trees based on the chloroplast DNA and ITS dataset, Panahi et al. concluded that there was reticulate evolution in section *Merwia*. They commented that this reticulate evolution resulted from hybridization and introgression, especially among Irano-Turanian species [26]. Therefore, the studies of the determination of putative new species based on the molecular data in the *Ferula* genus (especially in subgenus *Narthex*, section *Merwia*) are critical in revealing not only morphological but also phylogenetic evidence of the species and their infra-generic evolutionary relationships.

Specimens of the genus *Ferula* were collected by M. Miski and H. O. Tuncay from the shores of Tuz Lake (Konya Province) and Seyfe Lake (Kırşehir Province). Identification of the collected samples was attempted using the diagnostic key found in Flora of Turkey, as a result, the identified specimens were morphologically close to species *F. szowitsiana* DC. Thus, a detailed study was conducted, and it attempted to identify the collected species using the diagnostic keys in Flora of the U.S.S.R. and Flora Iranica [29,30]. Similarly, the collected specimens were found to be closely related to *F. szowitsiana* and *F. persica* species; however, they showed some differences from these two known species. Hence, more detailed comparative morphological, anatomical, chemical, and molecular studies were carried out to compare these potential new species with those of *F. szowitsiana* and *F. persica* species.

## 2. Results

### 2.1. Taxonomic Treatment

*Ferula turcica*: Akalın, Miski, and Tuncay sp. nov. (Figures 1 and 2).

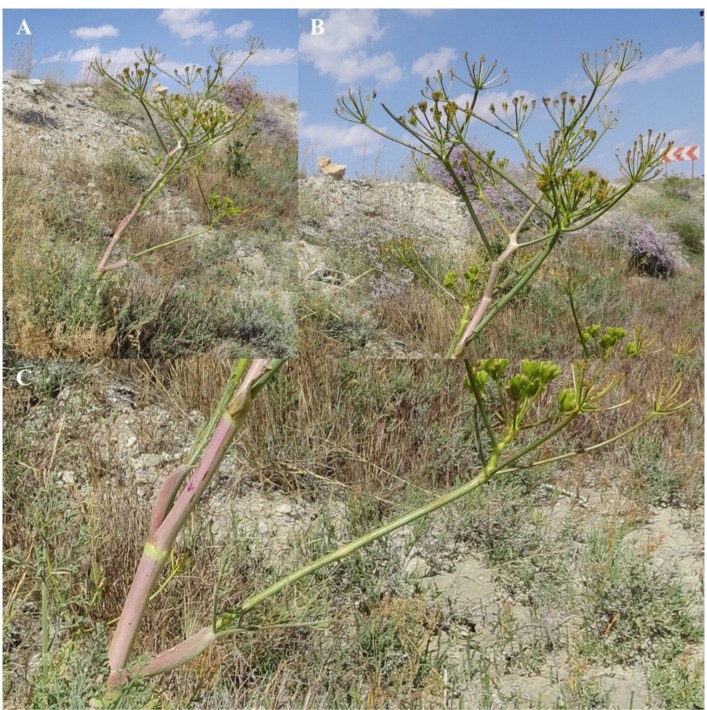

**Figure 1.** (**A**,**B**) General view of *Ferula turcica*. (**C**) Sheath.

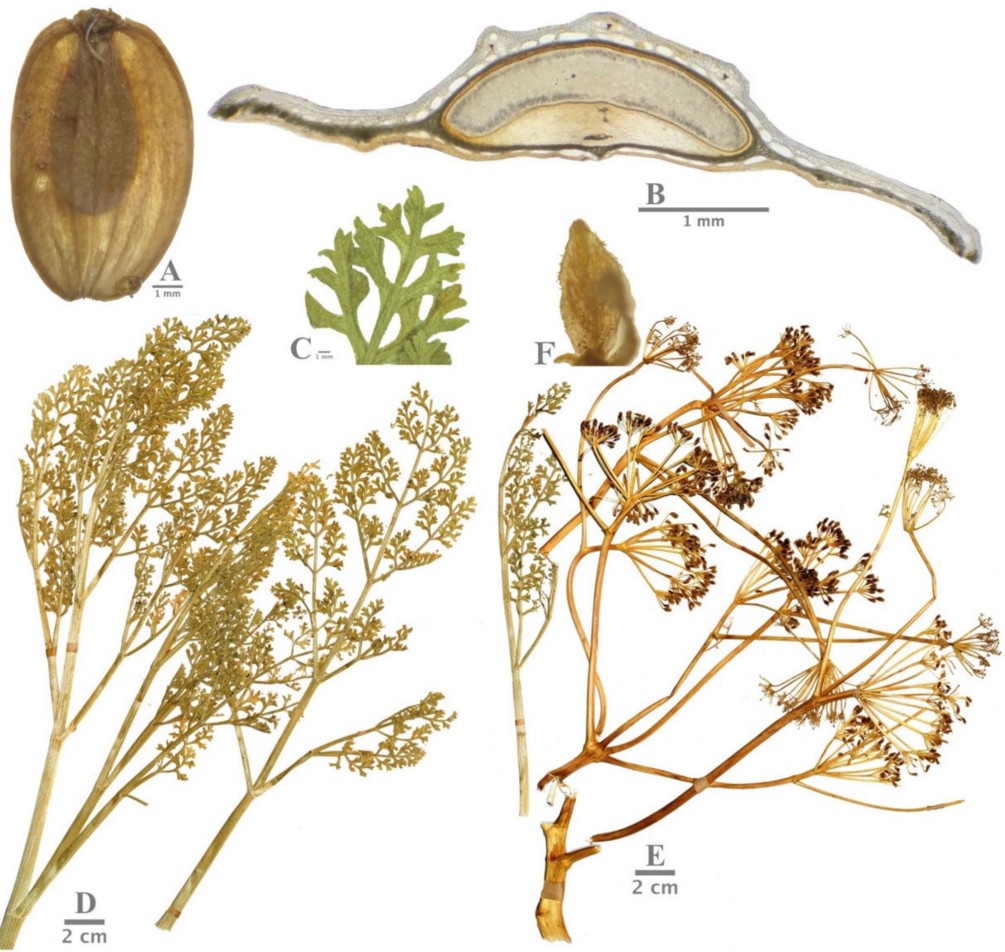

**Figure 2.** (**A**) General view of *Ferula turcica* fruit. (**B**) Cross-section of mericarp of *F. turcica*. (**C**,**D**) Basal leaf of *F. turcica*. (**E**) General view of *F. turcica* umbella. (**F**) Petals with setulose-puberulent hair.

Type: TÜRKİYE. B4 Konya: Tuz Lake, Yavşan salt pan, 910 m, 16 June 2015, M. Miski (Holotype: ISTE 116464).

Diagnosis: Caule crasso et striato, usque 100 cm; foliis basalia 5–6 pinnata, triangularia-ovata, 30–35 × 25–30 cm, dense pubescentia, segmentis ultimis pinnatisectis. Vaginae non in-flatae, 2 × 4–6 cm. Panicula laxa. Umbellis centralibus pedunculata 0.7–4 cm, umbellae lateralis raro solitaria vel 2–4 pedicellis longis, umbellis centralis (9-) 15–18 radiis 3–5 cm. Mericarpia elliptica, 5–7 mm × 9–11 mm, dorso dorsali leviter prominente et filiformi, alis lateralibus 1.2–1.9 mm latis, vittae dorsales 13–24, per valleculae 4–7 commissurales 11–14.

Description: Perennial herbs, erect, green, up to 100 cm tall, stem thick and striate. The root has a 1–3 cm width with a thick woody taproot system. A fibrous collar, which are old petioles, remains on the base of the stem. Leaves green, basal leaves 5–6 pinnate, triangular-ovate in outline, 30–35 × 25–30 cm, densely pubescent, ultimate segments pinnatisect, lobes 1–2.5 × 0.7–1 mm oblong, obtuse to acute. Sheats not inflated, 2 × 4–6 cm.

Inflorescence lax panicle, central umbels composed of fertile flowers, lateral umbels composed of sterile flowers. Central (terminal) umbella with peduncled 0.7–4 cm, lateral umbella rarely single or 2–4 on long pedicels, central umbella (9-) 15–18 rays 3–5 cm, umbellules (6-) 8–12 (−17) flowered; petals setulose-puberulent, pedicel at fruiting 0.5–1.4 mm long; sepals caducous in fruiting time.

Mericarps elliptic, 6.5 mm (5–7) width, 9.5 mm (9–11) length, dorsal ridges slightly protruding and filiform, lateral wings 1.5 mm (1.2–1.9) wide, dorsal vittae 13–24, 4–7 per vallecula, commissural 11–14 (Table 1).

**Table 1.** Comparison of the diagnostic characters of *Ferula turcica, F. latialata, F. szowitsiana,* and *F. persica*.

| Character | *F. turcica* | *F. latialata* | *F. szowitsiana* | *F. persica* |
|---|---|---|---|---|
| Stem | 70–100 cm | 80–110 cm | 50–70 cm | 70–100 cm |
| Ultimate segment of the leaf | Regular deeplobed 1–2.5 × 0.7–1 mm | Regular lobed 1–2 (−2.5) × 0.5–0.7 mm | Regular lobed 1–2 mm | Regular deeplobed |
| Hair | Densely puberulent | Puberulent | Setulose-puberulent | Pubescent |
| Ray numbers and length | (9-) 15–18 rays (3–5 cm) | 13–15 (−18) (3–5 cm) | 7–11 rays (2–5 cm) | 17–22 rays |
| Umbellules numbers | 8–12 (−17) | 8–10 (−14) | 8–12 | 15 |
| Pedicel at fruiting | 0.6–1.2 (−1.5) cm | 0.5–1 (−1.2) cm | 0.3–0.5 cm | - |
| Central umbella | Peduncled 0.7–4 cm | Peduncled 0.5–1.5 (−3) cm | Shortly peduncled or sessile | Sessile |
| Shape of fruit | Elliptic | Elliptic to oblong | Elliptic to orbicular | Ovoid |
| Lateral wings | 1.2–1.9 mm | 2.5–3.9 mm | 2–4 mm | - |
| Dorsal vittae | 4–7 per vallecula | 3–5 per vallecula | (2-) 4–6 per vallecula | 5–7 per vallecula |
| Commisural vittae | 11–14 | 6–10 | 8–12 | 16–18 |
| Width of fruit average (min-max) | 6 mm (5–7 mm) | 11 mm (9–12.5 mm) | 10–13 mm | 6 mm |
| Length of fruit average (min-max) | 9.5 mm (9–11 mm) | 15.5 mm (13.5–18 mm) | 12–20 mm | 11 mm |
| Ratio of length to width of fruit | 1.6 | 1.4 | 1.4 | 1.8 |

Etymology: *F. turcica* is named after the country of Türkiye.

Phenology: Flowering time is from May to June, and fruiting is from June to July.

*Ferula latialata*: Akalın, Miski, and Tuncay sp. nov. (Figures 3 and 4).

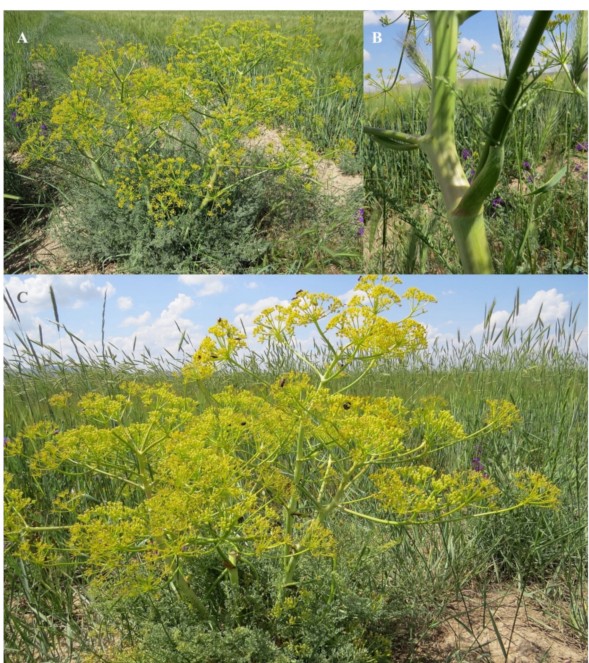

**Figure 3.** (**A**,**C**) General view of *Ferula latialata*. (**B**) Sheath.

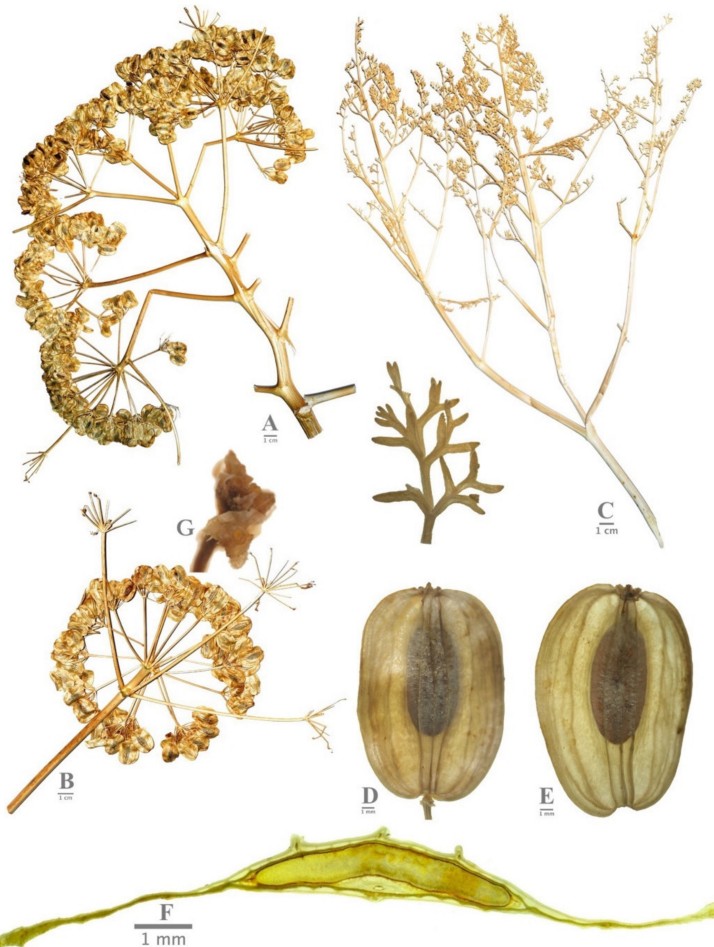

**Figure 4.** (**A**,**B**) General view of *Ferula latialata* umbella. (**C**) Basal leaf of *F. latialata*. (**D**,**E**) General view of *F. latialata* fruits. (**F**) Cross-section of mericarp of *F. latialata*. (**G**) Petals with setulose-puberulent hair.

Type: Türkiye. B5 Kırşehir: Seyfe Lake, near Yazıkınık village, 1110 m, 2 August 2021, H. O. Tuncay (Holotype: ISTE 117495).

Diagnosis: Caule crasso et striato, usque 110 cm; foliis basalia 6–7 pinnata, triangularia-ovata, 30–35 × 25–30 cm, pubescentia, segmentis ultimis pinnatisectis. Vaginae non inflatae, 2 × 4–6 cm. Panicula laxa. Umbellis centralibus pedunculata 0.5–1.5 (−2.5) cm, umbellae lateralis 1–3 (−4) pedicellis longis, umbellis centralis 13–15 (−18) radiis, 3–5 cm. Mericarpia elliptica vel oblonga, 13.5–18 mm × 9–12.5 mm, dorso dorsali leviter prominente et filiformi, alis lateralibus 2.5–3.9 mm latis, vittae dorsales per valleculae 3–5 commissurales 6–10.

Description: Perennial herbs, erect, green, up to 110 cm tall, stem thick and striate. Root 1–3 cm width with thick woody tap root system. A fibrous collar, which are old petioles, remains on the base of the stem. Leaves green, basal leaves 6–7 pinnate, triangular-ovate in outline, 30–35 × 25–30 cm, pubescent, ultimate segments pinnatisect, lobes 1–2 (−2.5) × 0.5–0.7 mm oblong, obtuse. Sheats not inflated, 2 × 4–6 cm.

Inflorescence lax panicle, central umbels composed of fertile flowers, lateral umbels composed of sterile flowers. Central (terminal) umbella with peduncled 0.5–1.5 (−2.5) cm, lateral umbella 1–3 (−4) on long pedicels, central umbella 13–15 (−18) rays 3–5 cm, umbellules 8–10 (−14) flowered; petals setulose-puberulent, pedicel at fruiting 0.5–1 mm long; sepals caducous in fruiting time.

Mericarps elliptic to oblong, 11 mm (9–12.5) width, 15.5 mm (13.5–18) length, dorsal ridges slightly protruding and filiform, lateral wings 3 mm (2.5–3.9) wide, dorsal vittae 3–5 per vallecula, commissural 6–10 (Table 1).

Etymology: The epithet name *latialata* from Latin, meaning wide, refers to the wide lateral wing in the fruit of *Ferula latialata*.

Phenology: Flowering time is from May to June, and fruiting is from June to July.

Distribution and ecology: The distribution of *Ferula turcica* (Konya) and *F. latialata* (Kırşehir) in Türkiye is shown in Figure 5. These two new species are distributed close to each other, and both are known from a single locality. Different localities have not yet been identified in field studies in similar habitats. *F. turcica* grows in halophytic soils near Tuz (salt) Lake, the second largest lake and an important source of salt in Türkiye, at an altitude of about 900 m, and its natural habitat is undisturbed. *F. turcica* grows together with *Ferula halophila* Peşmen, and they share the same habitat. The vegetation in the zone closest to the lake, which is covered with thick salt layers, consists of *Limonium lilacinum* (Boiss. and Balansa) Wagenitz, *Salicornia europaea* L., and *Halocnemum strobilaceum* Moris communities. The other new species, *F. latialata*, grows among the sunflower and wheat fields near Seyfe Lake, close to halophytic soils at an altitude of about 1100 m, and in relatively less saline soils than *F. turcica*. Species such as *Halocnemum strobilaceum* Moris, *Bassia Pilosa* (Fisch. and C.A. Mey.) Freitag and G. Kadereit, and *Camphorosma monspeliaca* L., are found in areas under the influence of the salty water of Seyfe Lake and salt marshes.

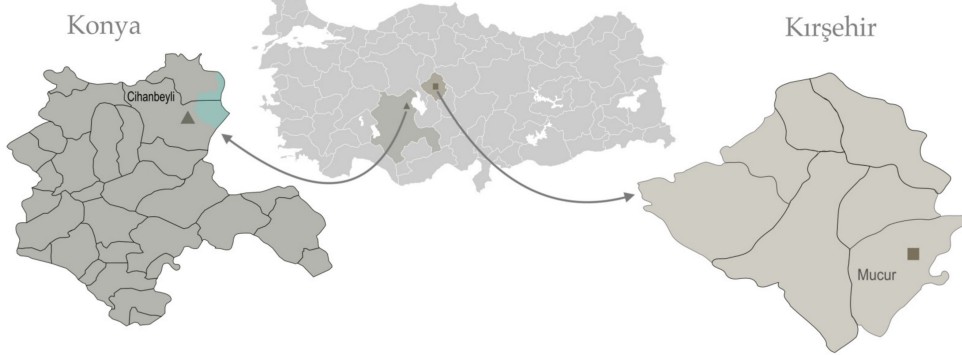

**Figure 5.** Distribution of *Ferula turcica* (Konya) and *F. latialata* (Kırşehir).

Conservation status: *Ferula turcica*, a halophyte, grows in rare habitats. Salt is extracted from Tuz Lake, where it grows, but the salt mine poses no threat to *F. turcica*. For this species, whose natural habitat is intact, its narrow distribution area and the rare soil structure in

which it grows may be limiting factors. The other new species, *F. latialata*, is more threatened than *F. turcica* because it grows in agricultural areas. The number of individuals observed in the population was low. The abandonment of agricultural areas may reduce the pressure on this species and increase the number of individuals in the population. Plant breeding studies have been started for *F. latialata*, which has high environmental pressure and a very limited number of individuals. In the present records, *F. turcica* and *F. latialata* are endemic taxon to central Anatolia and are known from only one locality; therefore, they are considered as "Endangered (criterion B1)". They could also be categorized Critically Endangered (criterion B2) due to their known area of occupancy of less than 2 km² and their population size estimated to be fewer than 250 mature individuals (Criterion C). It is suggested that the species of *F. turcica* and *F. latialata* should be considered Critically Endangered (CR) according to the IUCN threat criteria [31].

### 2.2. Phenetic Analysis

Principal coordinate analysis (PCoA) based on morphological, anatomical, and chemical data was performed on *Ferula turcica*, *F. latialata*, and related species (Figure 6). The 21 morphological and chemical characters used for the analysis are given in Table S1. Coordinate two clearly separates *F. turcica*, *F. latialata*, and *F. szowitsiana* from *F. drudeana* and *F. persica*, which are located on the positive side of the axis. Moreover, coordinate one separates *F. turcica* and *F. latialata* from allied species *F. szowitsiana*, which are located on the negative side of the axis.

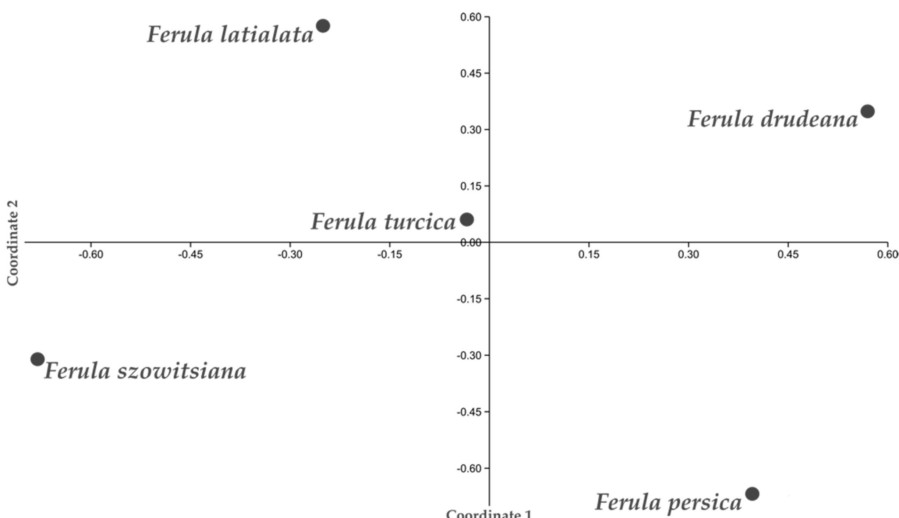

**Figure 6.** Principal coordinate analysis performed on 21 morphological, anatomical, and chemical characters of *Ferula turcica*, *F. latialata*, and related species.

### 2.3. Phylogenetic Evaluations

The phylogenetic hypothesis is presented in Figure 7. This tree is in agreement with those of Panahi et al. and Piwczyński et al. [26,27]. The genus *Ferula* is not monophyletic because of *Leutea*. The represented species of *Leutea* is nested in the *Ferula* ingroup in all the analyses, compatible with the results of Piwczyński et al. [27] and Panahi et al. [25,26]. The intra-generic classification in *Ferula* is not strongly supported in either subgenus nor are the sectional taxonomic levels by phylogenetic analysis. As discussed by Panahi, the species in nearly all the clades are located as polytomic terminals, maybe because of conspecific or nuclear-mediated introgression. Subgenus *Narthex* has numerous polytomic clades, which were determined as sections. The sampling is especially focused on subgenus *Narthex* section *Merwia* according to Panahi et al. [26]. Section *Merwia* does not generate a monophyletic clade. This section includes a clade (clade A) and two polytomic species, *F. kuhistanica* and *F. narthex*. Clade A is supported by medium posterior probability, whereas it is not supported by the bootstrap values (PP = 0.87). Clade A is divided into clades B and

C. Clade B is strongly supported by posterior probability, whereas it is not supported by the bootstrap values (PP = 0.93). This clade comprises numerous polytomic lineages. One of them, clade D, includes both of the new species, *F. turcica* and *F. latialata*. This clade is supported by strong posterior probability and moderate bootstrap values (PP = 0.99, BS-ML = 63, BS-MP = 80). *F. szowitsiana*, which was collected from the same locality (Tuzgölü, Yavşan salt pan) as *F. turcica* and is one of the accessions belonging to *F. drudeana*, are nested in clade D as polytomic lineages.

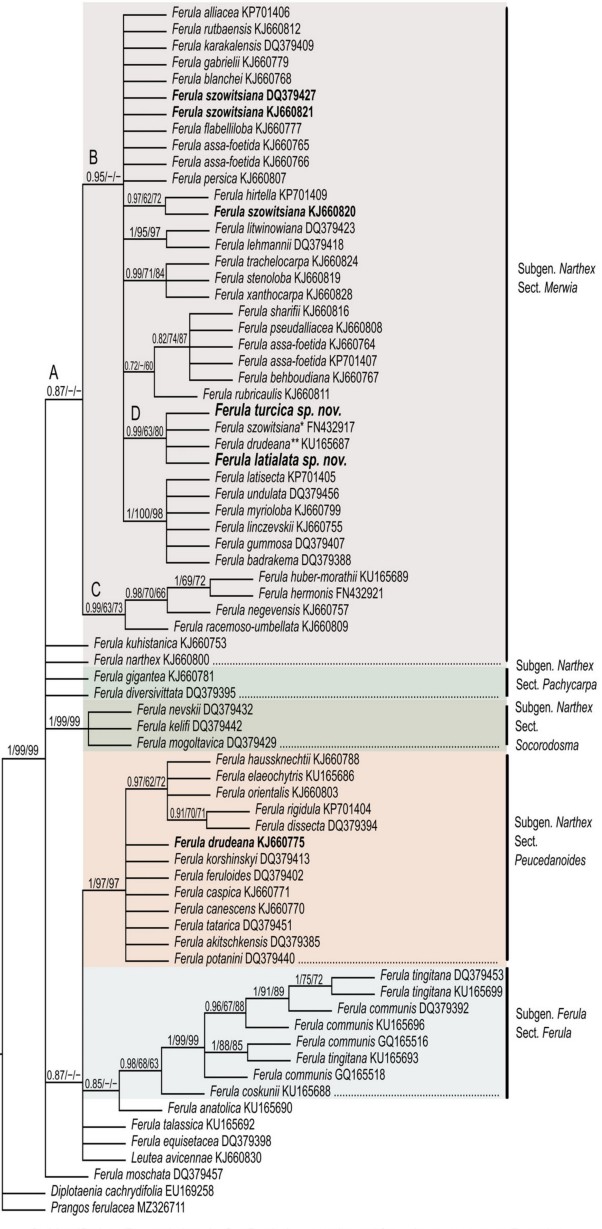

\* Identified as *F. szowitsiana* in GenBank. It was collected from the same as *F. turcica*.
\*\* In addition to being morphologically different from *F. turcica* and *F. latialata*, it is not in the same clade as the *F. drudeana* specimen collected by Siehe.

**Figure 7.** Bayesian estimate of the phylogeny of the genus *Ferula* focused on subgenus *Narthex* based on the ITS sequence dataset. The two new species described herein, *F. turcica* and *F. latialata*, and related species, *F. szowitsiana* and *F. drudeana*, are indicated by bold letters. The supporting posterior probability and bootstrap values (the first obtained from maximum likelihood analysis; the second obtained from maximum parsimony analysis) are presented above the branches. Alphabetical designations for some clades are addressed in the main text.

## 2.4. Chemotaxonomic Characteristics

The monograph of E. Korovin [14] lists 15 *Ferula* species in section *Merwia*, and the phytochemical investigations performed on some of these species indicated that their common secondary metabolites are sesquiterpene coumarins and sulfur-containing compounds [32–37]. Therefore, all *Ferula* species belonging to section *Merwia* in Türkiye have been classified as *F. szowitsiana*. However, preliminary phytochemical studies of *F. szowitsiana* collected from three locations in central Anatolia, i.e., the Kırşehir, Konya, and Sivas provinces, indicated that their phytochemical profiles are completely different. The major sesquiterpene coumarins isolated from the roots of *F. szowitsiana* were galbanic acid (**a**), methyl galbanate (**b**), and szowitsiacoumarin B (**c**) (Figure 8), rearranged monocyclic sesquiterpene coumarin ethers and bicyclic drimane type sesquiterpene coumarin ethers [38,39]. In contrast, the major sesquiterpene coumarins of the root extract of *Ferula turcica* collected from the shores of Tuz Lake were identified as kellerin (**d**), gummosin (**e**), persicasulphide A (**f**), and persicasulphide C (**g**), bicyclic drimane type sesquiterpene coumarin ethers and sulfur-containing compounds (Figure 8). Whereas colladonin (**h**), badrakemin (**i**), badrakemin acetate (**j**), and conferol (**k**), bicyclic drimane type sesquiterpene coumarin ethers, were found as the major compounds of *Ferula latialata* (Figure 8). Umbelliprenin (**l**), the biogenetic precursor of sesquiterpene coumarins of *Ferula* species, is also present in all three *Ferula* species (Figure 8). The HPLC chromatogram profiles (Figure S1) of the dichloromethane extracts of *F. latialata* (Kırşehir), *F. turcica* (Tuz Lake), and *F. szowitsiana* (Sivas) clearly showed that *F. latialata* and *F. turcica* should be different species individually.

**(a)** Galbanic acid
*F. szowitsiana*

**(b)** Methyl galbanate
*F. szowitsiana*

**(c)** Szowitsiacoumarin B
*F. szowitsiana*

**(d)** Kellerin
*F. turcica*

**(e)** Gummosin
*F. turcica*

**(f)** Persicasulphide A
*F. turcica*

**(g)** Persicasulphide C
*F. turcica*

**(h)** Colladonin
*F. latialata*

**(i)** Badrakemin
*F. latialata*

**(j)** Badrakemin acetate
*F. latialata*

**(k)** Conferol
*F. latialata*

**(l)** Umbelliprenin
*F. szowitsiana, F. turcica, F. latialata*

**Figure 8.** Secondary metabolites of *Ferula latialata*, *F. turcica*, and *F. szowitsiana*.

## 3. Discussion

*Ferula szowitsiana* is a species that is widely spread geographically, starting from Afghanistan, Turkmenistan, and Uzbekistan to Iran, Transcaucasus, and Türkiye. *F. persica*, which is quite close to *F. szowitsiana*, also spread in similar regions but not as widely as *F. szowitsiana*. The two new species, *F. turcica* and *F. latialata*, are morphologically close to *F. szowitsiana* and *F. persica*, which are members of subgenus *Narthex* section *Merwia* in Irano-Turanian groups. It was stated that the most prominent feature of subgenus *Narthex* is that the differentiation among the species of this subgenus is not clear [26]. When the two new species were compared to *F. szowitsiana* and *F. persica*, both new species were taller, had a greater number of rays, and had pedunculated central umbella. Compared to *F. latialata* and *F. turcica*, as the epithet name suggests, *F. latialata* has a wide lateral fruit wing and larger schizocarp fruit. In contrast, *F. turcica* has more dorsal and commissural vittae. Both new species grow in halophilous areas. It is thought that this habitat change may be one of the reasons for the differentiation from *F. szowitsiana*. The two new species, taxonomically supported by evident individual morphological, chemical, and molecular data, are considered not to be hybrids or variations of *F. szowitsiana* or *F. persica*.

The type specimen of *Ferula drudeana* was collected by Siehe (Siehe 408) [17]. Then, it was rediscovered by Sağıroğlu and Duman at the same location as the type specimen [40]. Sağıroğlu collected the specimens (Sağıroğlu 2525, GenBank barcode no: KU165687) from Kayseri. Another *F. drudeana* accession in GenBank (barcode no: KJ660775) was collected by Siehe (Siehe 163, in W). This population is nested in Section *Peucedanoides* (Figure 7, indicated by bold letters). *F. turcica* clearly differs from *F. drudeana* by puberulent petals (not glabrous), 4–7 vallecular vittae (not 2–3), and 1.2–1.9-mm wide lateral wings (not less than 0.5 mm). The other new species, *F. latialata*, has 2.5–3.9-mm wide lateral wings (not less than 0.5 mm) and setulose-puberulent petals (not glabrous). These distinct and significant morphological differences, supported by flower and fruit morphological characteristics, display that *F. drudeana* is different from both of the new species. Actually, *F. drudeana* is a remarkable species in the genus *Ferula* with its monocarpic form, thick stem, and glabrous petals. However, solving the polyphyly problem of *F. drudeana* needs more sampling and more morphological and phylogenetic studies. In clade D, the other species, *F. szowitsiana* (GenBank barcode no: FN432917), was collected from Tuzgölü, Yavşan salt pan. It is the type locality of *F. turcica*, as well. Additionally, more specimens from the Yavşan salt pan were collected by Davis (D.16670) and determined as *F. szowitsiana* [17]. On the other hand, *F. szowitsiana* was determined as a polyphyletic species (Figure 7, clade B), not only herein but also by Piwczyński et al. [27] and Panahi et al. [26]. It is clear that the Tuz gölü Yavşan populations identified as *F. szowitsiana* are genetically and morphologically different from the other collections of *F. szowitsiana* (Figure 7, clade B, Table 1). According to both our morphological and phylogenetic results, the new species could not be recognized, and therefore, these populations were misidentified as *F. szowitsiana*. As a result, the FN432917 specimen should be *F. turcica*, as well.

One of the major secondary metabolites from the root extracts of *Ferula turcica* species collected from the shores of Tuz Lake was previously isolated as the major secondary metabolite of *F. persica* [41,42]. However, the botanical features and phytochemical profile of *F. turcica* do not match those of *F. persica*. Moreover, the secondary metabolite profiles of *F. latialata* and *F. turcica* were not similar to those of *F. szowitsiana* and *F. persica*.

According to the data obtained, it is hypothesized that *Ferula persica* is at one end of the distribution of *F. szowitsiana*, and the two new species, *F. latialata* and *F. turcica*, are at the other end as a result of speciation. Aside from the observation of the morphological differences during the fieldwork, the chemical and molecular evidence supported the hypothesis that *F. turcica* and *F. latialata* are two new distinct species from *F. szowitsiana* and *F. persica*.

Identification key to *Ferula turcica, F. latialata*, and related species:

1. Leaf ultimate segment, linear-setaceous, dorsal vittae 2–3 per vallecula, petals glabrous
....................................................................................................................................................*drudeana*

1. Leaf ultimate segment, not linear-setaceous, dorsal vittae 3–7 per vallecula, petals setulose-puberulent.................................................................................................................2
2. Stem 50–70 cm, rays 7–11 per umbel.................................................................*szowitsiana*
2. Stem 70–110 cm, rays 11–22 per umbel..........................................................................3
3. Central umbella subsessile, mericarps ovoid, commissural vittae 16–18.....................*persica*
3. Central umbella peduncled, mericarps elliptic or elliptic to oblong, commissural vittae 6–14...........................................................................................................................4
4. Leaves densely puberulent, 15–18 rays, mericarps elliptic, lateral wings 1–2 mm ................................................................................................................................*turcica*
4. Leaves puberulent, 13–15 rays, mericarps elliptic to oblong, lateral wings 2.5–4 mm ................................................................................................................................*latialata*

## 4. Materials and Methods

The study was based on fieldwork, literature surveys, herbaria materials, and chemical and molecular studies. The new species materials were compared to the herbaria materials (Appendix A) of *Ferula* in Herbarium of Istanbul University Faculty of Pharmacy (ISTE), Royal Botanic Garden Edinburgh (E), Royal Botanic Gardens Kew (K), Moscow University Herbarium (MW), Natural History Museum Vienna (W), and Conservatoire et Jardin botaniques de la Ville de Genève (G). The voucher specimens were deposited in ISTE (*F. turcica* ISTE 116464, *F. latialata* ISTE 117495, *F. szowitsiana* ISTE116468).

Field studies were conducted by M. Miski and H.O. Tuncay in 2014, 2015, and 2021. *F. turcica* specimens were collected by M. Miski on 16 June 2015 from Konya. *F. latialata* specimens were collected from Kırşehir by M. Miski on 10 June 2014 and by H.O. Tuncay on 2 August 2021. Populations were observed during fieldwork, and a protection status was recommended according to IUCN threat criteria.

For fruit anatomy, the fruits were first submerged in a warm water–alcohol mixture (70% ethanol), and then all of the mericarps were cut by hand in the middle part with a razor. At least 40 mature fruits of *F. turcica and F. latialata* were analyzed. Samples were examined in Sartur reagent (a compound reagent of Sudan III, lactic acid, aniline, iodine, potassium iodide, water, and alcohol) [43]. Photographs were taken with an iPhone X. Measurements of the mericarps were made using ImageJ. The fruit morphology and anatomy were described using the terms of Botanical Latin [44] and Kızılarslan and Akalın [45]. The principal coordinate analysis, supported by the Gower similarity index, based on 21 morphological, anatomical, and chemical characters, was calculated and plotted with PAleontological STatistics (PAST) version 4.11 [46].

DNA extraction, amplification, sequencing, and phylogenetic analysis:

The total genomic DNAs were isolated from dried leaves of *F. turcica* and *F. latialata* using a GeneAll Plant SV Mini Kit (Seoul, Republic of Korea) following the protocol of the manufacturer. The complete ITS region in each genomic DNA was PCR-amplified using primers ITS4 and ITS5 [47]. Sequencing was performed by Atlas Biotechnologies (Ankara, Türkiye). The relevant marker sequences of all of the other taxa used for phylogenetic analysis were obtained from GenBank. Their NCBI barcode numbers are displayed in Figure 7. The outgroups were chosen according to Downie et al. and Panahi et al. [25,48]. ITS sequences (a total of two sequences) were obtained from the type population belonging to each of the two new species.

Representative species of the main lineages of the genus *Ferula* subgenus *Narthex* were selected according to the method of Panahi et al. and Piwczyński et al. [26,27] to determine the phylogenetic position of both new species. The aligned data matrix comprising 73 species × 618 alleles, was prepared using the MAFFT v. 7 online multi-alignment program [49] and analyzed with MrBayes v. 3.2.7a [50] with 20 million generations and a burn-in of 10% for the Bayes inference. The evolutionary substitution model was determined as TIM2+G using JModeltest 2.1.7 [51,52]. The ESS was checked using Tracer v. 1.7 [53]. The same dataset was analyzed using raxmlGUI v. 4.0b08 [54,55], with 500 runs and 1000 bootstrap replicates for the maximum likelihood approach. Additionally, the

maximum parsimony tree was calculated using MEGA11 based on the same dataset with 1000 bootstrap replications [56]. The phylogenetic trees were displayed and manipulated by FigTree v. 1.4.3 and TreeGraph 2 [57]. The posterior probability and bootstrap values (>50) were added to the branches of the Bayes tree in Figure 7.

Chemical Extraction:

Plant materials were extracted by maceration and continuous Soxhlet procedures, respectively. Dichloromethane (Merck, Darmstadt, Germany) was used during the extraction, and the solvent was evaporated by a rotary evaporator with low pressure and temperature. Once the dichloromethane extract was dissolved with acetone (Merck, Darmstadt, Germany) and allowed to rest at room temperature, it was kept at 4 °C until sedimentation of the hydrocarbons occurred. The sediment of the hydrocarbons was filtered by a Nuche Erlenmayer flask under a vacuum. The extract was then dried again using a rotary evaporator (Buchi, Flawil, Switzerland) with low temperature and pressure [58,59].

Analytical High-Performance Liquid Chromatography (HPLC):

Analytical HPLC analyses were performed using a Shimadzu 10A model apparatus (Shimadzu Analytical and Measuring Instruments, Kyoto, Japan). The system comprised a pump (LC-10AD), a diode-array detector (DAD) (SPD-M10A), and an autosampler (SIL-10AD). During the process, Shimadzu LC Solutions software was used to control the system and conduct post-run analyses of the data. Millipore (Billerica, MA, USA) was used to obtain Milli-Q ultrapure water. Water (ultrapure):acetonitrile (60:40 >> 0:100) solvents were used as the mobile phases with gradient elution, and a Luna C18 (Phenomenex, CA, USA) column was used as a stationary phase. The flow rate was 0.5 mL/min, the temperature was 30 °C, and the injection volume of the samples was 10 μL. The time of the analysis was arranged for 40 min, and the mobile phase was planned as follows: 0–5 min, 40% A and 60% B; 5–30 min, 0% A, 100% B; 30–40 min, 0% A, 100% B; 40–41 min, 40% A, 60% B. A wavelength of 259 nm was used during the analyses. The method was revised according to the extract to obtain quite a separation [60].

**Supplementary Materials:** The following supporting information can be downloaded at: https://www.mdpi.com/article/10.3390/horticulturae9020144/s1, Table S1: morphological, anatomical, and chemical characters used for principal coordinate analysis; Figure S1: comparison of (**A**) *Ferula latialata*, (**B**) *F. turcica*, and (**C**) *F. szowitsiana* HPLC chromatograms.

**Author Contributions:** Conceptualization, H.O.T., E.A. and M.M.; methodology, H.O.T., A.D.-K. and F.M.E.; software, H.O.T., A.D.-K. and F.M.E.; validation, H.O.T., E.A., A.D.-K., F.M.E. and M.M.; formal analysis, H.O.T., E.A., A.D.-K., F.M.E. and M.M.; investigation, H.O.T., E.A., A.D.-K., F.M.E. and M.M.; resources, H.O.T., E.A., A.D.-K., F.M.E. and M.M.; data curation, H.O.T., E.A., A.D.-K., F.M.E. and M.M.; writing—original draft preparation, H.O.T., A.D.-K. and F.M.E.; writing—review and editing, H.O.T., E.A., A.D.-K., F.M.E. and M.M.; visualization, H.O.T., A.D.-K. and F.M.E.; supervision, E.A., M.M. and A.D.-K.; project administration, H.O.T., E.A., A.D.-K., F.M.E. and M.M.; funding acquisition, H.O.T., E.A., A.D.-K., F.M.E. and M.M. All authors have read and agreed to the published version of the manuscript.

**Funding:** This study was funded by the Scientific Research Projects Coordination Unit of Istanbul University. Project number: 30731.

**Data Availability Statement:** Not applicable.

**Conflicts of Interest:** The authors declare no conflict of interest.

## Appendix A

Additional herbarium specimens examined morphologically:

*Ferula szowitsiana*: ISTE 109444, ISTE 109420, ISTE 73511, ISTE 19932, ISTE 15377, ISTE 105388, ISTE 12865, ISTE 21102, ISTE 62970; E00428291, E00428292, E00392437, E00175642, E00175641, E00175640, E00433774, E00433776, E00467554, E00467555, E00467556, E00467557, E00467558, E00467559, E00467560, E00467560, E00467562, E00467563, E00467564, E00467565, E00467566,

E00467567, E00467568, E00467569, E00467570, E00467571, E00467572, E00467573, E00467574, E00467575, E00279094; K001097219, K001097218, K000568185; MW0744692, MW0744691.

*Ferula persica*: E00205681, E00467654, E00467655, E00467656, E00360731; K001097211; MW0744674, MW0744675, MW0744676, MW0744677, MW0744678, MW0744679, MW0700492, MW0700493, MW0754156.

*Ferula persica* var. latisecta: W1961-0001614.

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
