# Peer review of "Two New Ferula (Apiaceae) Species from Central Anatolia: Ferula turcica and Ferula latialata"

_horticulturae, doi:10.3390/horticulturae9020144_

Round 1

Reviewer 1 Report

Thanks for your invitation for reviewing this paper, which proposes two new plant species of Ferula, named F. turcica and F. latialata. Authors provided nice morphological characters using figures. However, I am afraid that the manuscript is not suitable to be published in the journal Horticulturae (a very good journal in Plant Science). Firstly, authors need to re-construct the phylogram using MP, ML and Bayes methods, and show the difference of the sequence data in the manuscript. At present version, we cann’t see the difference of two new species from the phylogenetically-close species, which cann’t support the result of the establishment of the two new species. Hence, this is important to submit the original sequence data of the new species for review. Secondly, there are many writing mistakes in this manuscript, and authors need to carefully to revise them. Hope the above are helpful to improve this paper.

I am not familiar with this genus, however, the proposal for these two species seems reasonable.

Author Response

We've carefully re-examined the entire manuscript. Your criticisms have been very helpful. Thank you very much. If the manuscript needs specific correction we are ready. Please see the attachment.

Reviewer 2 Report

Suggestions and recommendations: 

row 16: correct "found" to "discovered"; row 100: before "910 m" insert coordinates N,E for the locus classicus; row 122: before "Flowering" add: "Phenology:", correct "in fruit" to "fruiting"; after row 121 add following: "Etymology:" and insert text from the row 166; row 130: before "1110 m" insert coordinates N,E for the locus classicus; after row 151 add "Etymology" and insert text from the row 167-168; Row 152: before "Flowering" add: "Phenology:", correct "in fruit" to "fruiting"; Table 1, first column, correct title "Characters" to "Character"; row 176: Subtitle: "Recommended IUCN threat category;" change to "Conservation status; row 181-182, part of the sentence "should be placed..." change to: "should be condidered as Critically Endangered (CR) according to the IUCN threat criteria"; row 199: please add more information why the genus Ferula is not monophyletic because of Leutea; Chapter References needs substanital corrections by following the Instructions for Authors - 33 out of total 56 references (59 %) are not correctly written, tje journal names must be abbreviated.

The topic is of great botanical interest because it delivers the new knowledge on diversity of genus Ferula, and description of two new species is a major achievements for the authors. The methodology is up to date, and reflects the phylogeny and taxonomic relationship in the most accurate way. 

The authors are encouraged to improve the manuscript. 

Author Response

(The authors gave the same response as above.)

Reviewer 3 Report

Dear Authors,  

I have reviewed the manuscript carefully, there are major gaps to be fixed.

I would suggest improving the formatting and layout of the manuscript. 

The part that needs most adjustment is the materials and methods part, it is poorly understood in some places and with big gaps in others.

The part on the distribution and ecology of the two proposed new species also needs to be fixed and expanded.  

Find all detailed comments and suggestions for each part in the appendix.  

The manuscript in general is well done and is very interesting, with the suggested changes it could become a really good article.  

Author Response

(The authors gave the same response as above.)

Round 2

Reviewer 1 Report

Authors have improved the phylogram, and discussed the raltionships among the related species. However, please further check the language.

Author Response

Your criticisms have been very helpful. Thank you very much. Manuscript has been edited for English language and grammar as well as scientific content and formatting. Please find the attachment.

Reviewer 3 Report

Chapter references; some references are not written correctly.

Ecology of the two species can be written better, same with regard to conservation status.

Figure 9 can go in the supplements to lighten the text.

The manuscript has been greatly improved, especially the materials and methods part. 

Now it is really a very good manuscript.

Author Response

(The authors gave the same response as above.)
